# A Calculation Model of the Dimensionless Productivity Index Based on Non-Piston Leading Edge Propulsion Theory in Multiple Oilfield Development Phases

**Bin Huang** [1,2,*]**, Qiaoyue Liang** [1]**, Cheng Fu** [1,2,*]**, Chunbai He** [3]**, Kaoping Song** [1] **and Jinzi Liu** [4]

[1]   Key Laboratory of Enhanced Oil Recovery (Northeast Petroleum University), Ministry of Education, College of Petroleum Engineering, Northeast Petroleum University, Daqing 163318, China; lqyemail@126.com (Q.L.); skpskp01@sina.com (K.S.)

[2]   Post-Doctoral Scientific Research Station, Daqing Oilfield Company, Daqing 163413, China

[3]   CNOOC Research Center CNOOC Research Institute, Beijing 10027, China; hechb@cnooc.com.cn

[4]   School of Mathematics and Statistics, Northeast Petroleum University, Daqing 163318, China; jinzi19811216@126.com

*   Correspondence: huangbin111@163.com (B.H.); cheng_fu111@163.com (C.F.)

**Abstract:** The dimensionless productivity index is an important indicator for measuring the oil production capacity of oilfields. The traditional calculation method of the dimensionless productivity index is not suitable for the continuous multiple development phases of oilfields. In this study, based on Darcy's Law and the theory of non-piston leading edge propulsion, we considered the influence of capillary pressure and derived a differential equation for leading edge propulsion distance. We established a calculation model of the dimensionless productivity index that is suitable for the multiple development phases of oilfields, including water flooding, polymer flooding, and binary compound flooding. The model was applied to the W block of the JZ9-3 oilfield, and the calculation results of the model were compared with the actual statistical results. The results show that the calculation error rates of the dimensionless productivity index in three phases of oilfield development are 4.67%, 17.65%, and 18.50%, respectively, and the average error rate is 10.38% in the overall development phase. The dimensionless productivity index curve shows a trend of first rising, then falling, and finally stabilizing when the pore volume number is included. This calculation model expands the field application scope of the theoretical dimensionless productivity index, which is convenient for application in oilfields, and improves the efficiency of the comprehensive evaluation of oilfields during multiple development phases.

**Keywords:** non-piston leading edge propulsion theory; the dimensionless productivity index; capillary pressure; polymer flooding; binary compound flooding

## 1. Introduction

The dimensionless productivity index is the ratio of liquid production to oil production during the initial anhydrous production period of an oilfield. This indicator directly reflects the variation law of liquid production and can be used to analyze the development effect and predict the actual liquid production capacity of oilfields. At present, the three main research methods used for the dimensionless productivity index are: reservoir field data calculation, numerical simulation, and theoretical calculation of seepage mechanics.

Chen and Zhang [1] first proposed that by calculating the liquid yield index from the actual reservoir field data, the dimensionless productivity index can be changed. Diyashev [2] proposed

calculating the dimensionless productivity index based on field measurement production, flow pressure, and well and reservoir data to determine the effectiveness of production and whether production improvement measures are required. Albertus [3] proposed a generalized dimensionless productivity index model based on nonlinear regression analysis of simulation results. The performance of multi-branch wells in the reservoir can be predicted, this method has limitations when applied to two-phase reservoir flow. Yu et al. [4] established a heterogeneous numerical model of the development of fluvial reservoirs under the condition of flat water flooding. The orthogonal design method was used to numerically simulate 27 groups of schemes, and the multivariate regression method was used to obtain the dimensionless productivity index. Perna [5] used the dimensionless productivity index to define the extent of reservoir damage, the effectiveness of stimulation measures, the shortest run time between operations, and sudden changes in well condition, using interpretation and validation through processing and interpretation of raw data. Li et al. [6] considered the influence of the well network on the dimensionless productivity index and verified their findings using the Xingbei development zone oilfield as an example. Garnica et al. [7] considered the influence of reducing the viscosity on the dimensionless productivity index and verified their findings using numerical simulation. Guo [8] examined the influence of the average viscosity of oil–water two-phase flow zone on the dimensionless productivity index and verified their findings using numerical simulation. The formula of dimensionless productivity index before and after borehole water breakthrough was deduced. Yang [9] conducted mathematical analysis and multiple numerical simulation on the Corey type function based on the Buckley–Leverett (B–L) equation and relative permeability, and derived a new model for predicting the dimensionless productivity index. Kaul [10] proposed the introduction of dimensionless fracture conductivity into a dual porosity model based on the dimensionless productivity index and its derivatives, revealing the supporting effect and influence of anti-drainage on the long-term performance of hydraulic fracturing horizontal wells. Hashmi [11] summarized the law of change in the dimensionless productivity index with dimensionless time, and defined the propagation area of the dimensionless average reservoir pressure in the reservoir, overcoming the shortcoming that the velocity transient analysis method cannot decouple the permeability from the flow field of the dual porosity reservoir. Using the relative permeability curve, Jiang et al. [12] deduced the formula of dimensionless productivity index under different water conditions without considering the change in fluid properties with pressure. This formula can be used to predict the liquid supply capacity of a certain small layer. Poe Jr. [13] established a solution method for the dimensionless productivity index of closed rectangular boundary reservoirs. Zhu [14] used a newly developed distributed volume source (DVS) method to calculate the dimensionless productivity index defined in a box-like region. This method is suitable for transient flow and quasi-steady flow. Medeiros [15] concluded that the dimensionless productivity index showed a sharp downward trend, analyzed the hydraulic fracturing production data of horizontal wells, and numerically simulated the natural fracture region. Li [16] deduced the formula for calculating the dimensionless productivity index of low permeability reservoirs considering capillary pressure. According to the characteristics of the flat section on the dimensionless productivity index curve, the difference in production performance between high and low permeability oil wells were explained. Gu et al. [17,18] first described the production characteristics of low-permeability oilfields, and reported that considering the differences in the capillary force and gravity would increase or decrease the dimensionless productivity index. Therefore, the yield stress of water and the capillary force were ignored in the formula derivation. Then, they considered the change in the water saturation in injection-production wells. Through the deformation treatment of the B–L equation in the water flooding after seeing the water, a new method for calculating the dimensionless productivity index was deduced. Papoutsidakis [19] proposed Input to State Stability (ISS) control, and considered how the method could be used by a scientifically-oriented engineer, either for human measuring or in production. Their study assumed an interdisciplinary perspective in the evaluation of oilfield production.

All of the studies reviewed above lack a comprehensive comparative analysis of the continuous multiple development phases of oilfields. Although the calculation methods of the dimensionless productivity index according to the reservoir field data are more accurate, the influences of different factors on fluid production changes cannot be clearly described. The production pressure differences of different water-bearing stages need to be statistically analyzed, and the processing is cumbersome. Numerical simulation methods require a large amount of data, which complicates the fitting of the formation pressure, are basically applied to the development stage of water flooding, and lack the evaluation of the liquid production level in the chemical flooding development stage. At present, the research methods of the change rule of the dimensionless productivity index based on the theory of seepage mechanics are mostly based on the water flooding development stages of single-layer and single wells. The methods for calculating the dimensionless productivity index of the polymer flooding stage is studied using water flooding, and the reservoir field data show that the calculation results of the dimensionless productivity index in polymer flooding stage are significantly different from the theoretical value.

After comprehensive consideration of the above factors, to study the variation law of reservoir production capacity in multi-development phases of oilfield, this study is based on the theory of non-piston leading edge propulsion under considering capillary pressure. At constant pressure, we established the dimensionless productivity index calculation models of water flooding, polymer flooding, and binary compound flooding. We applied the models to different development stages of the W block of the JZ9-3 oilfield (Tianjin, China). The JZ9-3 reservoir is an offshore heavy reservoir and is characterized by high reservoir permeability, large effective thickness, strong heterogeneity in the intra-layer, and large water–oil viscosity ratio. The W block of the oilfield has been in production since October 2000, and full injection polymer occurred in February 2008. To enhance the oil displacement effect, surfactant was added to the injection in 2010, and the injection well entered into the binary compound flooding stage in June 2011. Therefore, the JZ9-3 oilfield can be used to verify the feasibility of the calculation model and for comparison with the actual data provided by the field data to verify accuracy of the calculation method, providing guidance for later oilfield development.

## 2. Establishment of a Calculation Model of the Dimensionless Productivity Index

The presence of capillary pressure, the oil–water two-phase flow in the formation fluid, presents a non-piston leading edge propulsion trend, and the leading edge of displacement is prone to forming an oil-rich zone. The seepage resistance of the formation fluid increases obviously, and causes a substantial decrease in liquid production in some wells and a decline in the degree of production compared with the water drive stage. Therefore, based on the traditional calculation model of the dimensionless productivity index, we considered the influence of capillary pressure, and derived the formula for calculating the dimensionless productivity index in the water flooding-polymer flooding-binary compound flooding stages. The following details the whole process of formula derivation.

### 2.1. Formula for Leading Edge Propulsion Distance

In the calculation of dimensionless productivity index, the amount of liquid production at the initial stage should be solved first. At the initial stage, the formation is the single-phase flow of oil. According to Darcy's Law, the calculation formula of the output at this time can be obtained [20,21]:

$$J_D = \frac{Q_t}{Q_o} \tag{1}$$

$$Q_o = \frac{k_o A \Delta p}{\mu_o L} = \frac{k_{ro} K A \Delta p}{\mu_o L} \tag{2}$$

$$Q_w = \frac{k_w A \Delta p}{\mu_w L} = \frac{k_{rw} K A \Delta p}{\mu_w L} \tag{3}$$

$$Q_t = Q_o + Q_w = \frac{k_{ro}KA\Delta p}{\mu_o L} + \frac{k_{rw}KA\Delta p}{\mu_w L} \tag{4}$$

where $J_D$ is the dimensionless productivity index, $Q_t$ is the liquid production (m³/s), $Q_o$ is oil production (m³/s), $Q_w$ is the water production (m³/s), $\mu_o$ is oil phase viscosity (mPa·s), $\mu_w$ is the water phase viscosity (mPa·s), $K_{ro}$ is the oil phase relative permeability ($10^{-3}$ μm²), $K_{rw}$ is the relative permeability of the water phase ($10^{-3}$ μm²), $K_o$ is the oil phase permeability ($10^{-3}$ μm²), $K_w$ is the water phase permeability ($10^{-3}$ μm²), $K$ is the reservoir permeability ($10^{-3}$ μm²), $P$ is the pressure difference along the displacement direction (MPa), $L$ is the band length (m), $\Delta P$ is the pressure difference (MPa), and $A$ is the sectional area (m³).

The calculation formula of leading edge advancing distance in each stage is:

$$x = \int \frac{v_t}{\phi} \frac{df_w}{ds_w} dt \tag{5}$$

where $\phi$ is the porosity (%), $x$ is the leading edge advancing radius (m), $f_w$ is the water cut (%), $S_w$ is the water saturation (%), and $v_t$ is the seepage velocity of the total liquid yield (m/s).

## 2.2. Calculation Formula of Water Flooding-Polymer Flooding-Binary Compound Flooding in Multi-Development Phases

Because of the different water–oil viscosity ratios during different stages, causing the leading edge advancement are different. Therefore, the calculation models of the dimensionless productivity index in the stages of water flooding, polymer flooding, and binary compound flooding are specifically discussed.

Since gravity has little effect on the dimensionless productivity index, the influence of gravity is ignored here. According to the multi-phase flow seepage mechanism, the oil–water two-phase Darcy's law motion equations shown in Equations (6) and (7) are obtained:

$$v_w = -\frac{K_w}{\mu_w} \frac{\partial p_w}{\partial x} \tag{6}$$

$$v_o = -\frac{K_o}{\mu_o} \frac{\partial p_o}{\partial x} \tag{7}$$

where $v_w$ is the water phase migration velocity (m/s) and $v_o$ is the oil phase migration velocity (m/s).

The seepage velocity of the total liquid yield is:

$$v_t = v_o + v_w = \frac{K_w}{\mu_w} \frac{\partial p_w}{\partial x} + \frac{K_o}{\mu_o} \frac{\partial p_o}{\partial x} \tag{8}$$

The calculation formula for moisture content is:

$$f_w = \frac{v_w}{v_t} \tag{9}$$

Equations (6)–(9) can be obtained by considering the moisture content splitting equation under the influence of capillary pressure:

$$f_w = \frac{1 + \frac{K_o}{\mu_o} \frac{1}{v_t} \frac{\partial p_c}{\partial x}}{1 + \frac{\mu_w}{\mu_o} \frac{K_o}{K_w}} \tag{10}$$

The moisture content considering the capillary pressure is greater than the moisture content without considering the capillary pressure. Among them,

$$\frac{\partial p_c}{\partial x} = \frac{\partial p_o}{\partial x} + \frac{\partial p_w}{\partial x} \tag{11}$$

where $P_c$ is capillary pressure (Mpa), and $P_o$ and $P_w$ are oil and water phase pressure, respectively ($10^{-1}$ Mpa).

$K_o$ and $K_w$ are $f_w$-related functions, which can be expressed as:

$$K_o = K \cdot K_{ro}(S_w) \tag{12}$$

$$K_w = K \cdot K_{rw}(S_w) \tag{13}$$

According to Equations (10)–(12), the moisture content $f_w$ is obtained by deriving the water saturation $S_w$:

$$\frac{df_w}{ds_w} = \frac{\frac{K}{\mu_o}\frac{1}{v_t}\frac{\partial p_c}{\partial x}(K_{ro})'\left(1 + \frac{\mu_w}{\mu_o}\frac{K_o}{K_w}\right) - \left(1 + \frac{K_o}{\mu_o}\frac{1}{v_t}\frac{\partial p_c}{\partial x}\right)\frac{\mu_w}{\mu_o}\left(\frac{K_{ro}}{K_{rw}}\right)'}{\left(1 + \frac{\mu_w}{\mu_o}\frac{K_o}{K_w}\right)^2} \tag{14}$$

The seepage velocity of the total liquid production can also be expressed as:

$$v_t = \frac{q(t)}{A} \tag{15}$$

Substituting Equations (14) and (15) into Equation (5), we obtain the differential equation of non-piston leading edge propulsion position under the action of capillary pressure:

$$x = \frac{1}{A\phi}\left(\frac{\frac{K}{\mu_o}\frac{\partial p_c}{\partial x}(K_{ro})'}{1 + \frac{\mu_w}{\mu_o}\frac{K_o}{K_w}} - \frac{\frac{K_o}{\mu_o}\frac{\partial p_c}{\partial x}\frac{\mu_w}{\mu_o}\left(\frac{K_{ro}}{K_{rw}}\right)'}{\left(1 + \frac{\mu_w}{\mu_o}\frac{K_o}{K_w}\right)^2} - \frac{\frac{\mu_w}{\mu_o}\left(\frac{K_{ro}}{K_{rw}}\right)'}{\left(1 + \frac{\mu_w}{\mu_o}\frac{K_o}{K_w}\right)^2}\right)\int_0^t q(t)dt \tag{16}$$

where $\int_0^t q(t)dt = W_i$ is the total liquid production rate (m$^3$).

Based on the above Darcy's Law and non-piston leading edge propulsion theory, the dimensionless productivity index variation model is deduced, the concept of leading edge radius is introduced, and the differential equation of leading edge propulsion position under capillary pressure is established. The mathematical model is suitable for water flooding, polymer flooding, and binary compound flooding

### 2.2.1. Water Flooding Stage

As shown in Figure 1(a), the amount of liquid production during the water flooding stage is:

$$Q_{t1} = \frac{k_{ro}KA\Delta p}{(L-x)\mu_o} + \frac{KA\Delta p}{x}\left(\frac{k_{ro}}{\mu_o} + \frac{k_{rw}}{\mu_w}\right) \tag{17}$$

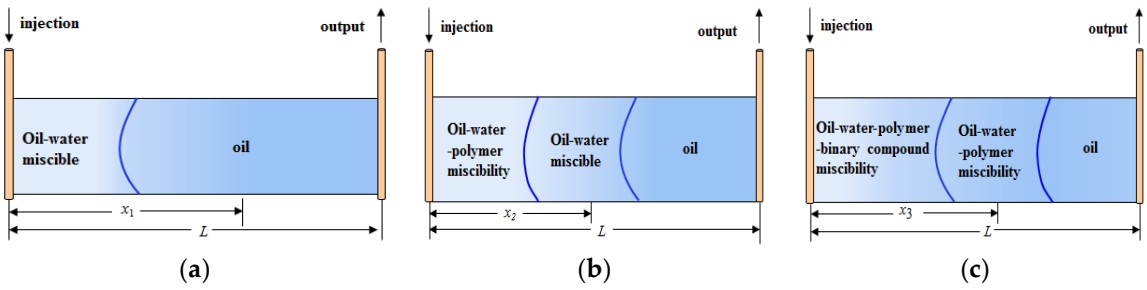

**Figure 1.** Schematic diagram of the leading edge propulsion of (**a**) water flooding, (**b**) polymer flooding, and (**c**) binary compound flooding stages in the multi-development phases of an oilfield.

Substituting Equations (2) and (17) into Equation (1) to obtain the dimensionless productivity index of the water flooding stage, we obtain:

$$J_{D1} = \frac{\frac{L}{x}\left(\frac{k_{ro}}{\mu_o} + \frac{k_{rw}}{\mu_w}\right) + \left(\frac{L}{L-x}\right)\frac{k_{ro}}{\mu_o}}{\frac{k_{ro}}{\mu_o}} \tag{18}$$

When $x_1$ is the leading edge radius of the water flooding stage:

$$x_1 = \frac{1}{A\phi}\left(\frac{\frac{K}{\mu_o}\frac{\partial p_c}{\partial x}(K_{ro})'}{1 + \frac{\mu_w}{\mu_o}\frac{K_o}{K_w}} - \frac{\frac{K_o}{\mu_o}\frac{\partial p_c}{\partial x}\frac{\mu_w}{\mu_o}\left(\frac{K_{ro}}{K_{rw}}\right)'}{\left(1 + \frac{\mu_w}{\mu_o}\frac{K_o}{K_w}\right)^2} - \frac{\frac{\mu_w}{\mu_o}\left(\frac{K_{ro}}{K_{rw}}\right)'}{\left(1 + \frac{\mu_w}{\mu_o}\frac{K_o}{K_w}\right)^2}\right)\int_0^t q_1(t)dt \tag{19}$$

where $\mu_w$ is the aqueous phase viscosity (mPa·s) and $K_{ro}/K_{rw}$ is the fluid-to-water ratio.

### 2.2.2. Polymer Flooding Stage

As shown in Figure 1(b), the Polymer solutions are high viscosity non-Newtonian fluids, with viscosity greater than aqueous phase viscosity. After polymer injection, a heavy oil zone easily forms at the displacement front, and the migration resistance of formation fluid is significantly improved, leading to a decline in the liquid production capacity of some wells compared with the water flooding stage [10].

The amount of liquid produced during the polymer flooding phase is:

$$Q_{t2} = \frac{KA\Delta p}{L - x_p}\left(\frac{k_{ro}}{\mu_o} + \frac{k_{rw}}{\mu_w}\right) + \frac{k_{rp}KA\Delta p}{x_p\mu_p} \tag{20}$$

Substituting Equations (2) and (20) into Equation (1) to obtain the dimensionless productivity index of the polymer flooding stage, we obtain:

$$J_{D2} = \frac{\frac{L}{x_p}\frac{k_{rp}}{\mu_p} + \frac{L}{L-x_2}\left(\frac{k_{ro}}{\mu_o} + \frac{k_{rw}}{\mu_w}\right)}{\frac{k_{ro}}{\mu_o}} \tag{21}$$

When $x_2$ is the leading edge radius of the polymer flooding state,

$$x_2 = \frac{1}{A\phi}\left(\frac{\frac{K}{\mu_o}\frac{\partial p_c}{\partial x}(K_{ro})'}{1 + \frac{\mu_p}{\mu_o}\frac{K_o}{K_p}} - \frac{\frac{K_o}{\mu_o}\frac{\partial p_c}{\partial x}\frac{\mu_p}{\mu_o}\left(\frac{K_{ro}}{K_{rp}}\right)'}{\left(1 + \frac{\mu_p}{\mu_o}\frac{K_o}{K_p}\right)^2} - \frac{\frac{\mu_p}{\mu_o}\left(\frac{K_{ro}}{K_{rp}}\right)'}{\left(1 + \frac{\mu_p}{\mu_o}\frac{K_o}{K_p}\right)^2}\right)\int_0^t q_2(t)dt \tag{22}$$

where $\mu_w$ is the polymer phase viscosity (mPa·s).

### 2.2.3. Binary Compound Flooding Stage

As shown in Figure 1(c), under the condition of binary compound flooding, the introduction of surfactant as oil displacement agent leads to complex physical and chemical reactions between oil and water, and also affects the liquid production capacity of oil wells. Therefore, studying the variation rule of liquid production capacity in chemical flooding to maintain high and stable production in heavy oil reservoirs is crucial [10].

The amount of liquid production during the binary compound flooding stage is:

$$Q_{t3} = \frac{k_{re}KA\Delta p}{x_3\mu_e} + \frac{KA\Delta p}{L - x_3}\left(\frac{k_{ro}}{\mu_o} + \frac{k_{rw}}{\mu_w} + \frac{k_{rp}}{\mu_p}\right) \tag{23}$$

Substituting Equations (2) and (23) into Equation (1) to obtain the dimensionless productivity index of the binary compound flooding stage, we have:

$$J_{D3} = \frac{\frac{L}{x_3}\frac{k_{re}}{\mu_e} + \frac{L}{L-x_3}\left(\frac{k_{ro}}{\mu_o} + \frac{k_{rw}}{\mu_w} + \frac{k_{rp}}{\mu_p}\right)}{\frac{k_{ro}}{\mu_o}} \tag{24}$$

When $x_3$ is the leading edge radius of the binary compound flooding stage,

$$x_3 = \frac{1}{A\phi}\left(\frac{\frac{K}{\mu_o}\frac{\partial p_c}{\partial x}(K_{ro})'}{1 + \frac{\mu_r}{\mu_o}\frac{K_o}{K_r}} - \frac{\frac{K_o}{\mu_o}\frac{\partial p_c}{\partial x}\frac{\mu_r}{\mu_o}\left(\frac{K_{ro}}{K_{rr}}\right)'}{\left(1 + \frac{\mu_r}{\mu_o}\frac{K_o}{K_r}\right)^2} - \frac{\frac{\mu_r}{\mu_o}\left(\frac{K_{ro}}{K_{rr}}\right)'}{\left(1 + \frac{\mu_r}{\mu_o}\frac{K_o}{K_r}\right)^2}\right)\int_0^t q_3(t)dt \tag{25}$$

where $\mu_r$ is the binary phase viscosity ( mPa·s) and $K_{ro}/K_{rr}$ is the oil binary mobility ratio.

From the above derivation, the dimensionless productivity index is related to the rock permeability, water saturation, the water–oil viscosity ratio, and the capillary pressure gradient. Based on the study of the differential equation of the leading edge propulsion distance at each stage of development, the dimensionless productivity index variation rule of the water-flooding-polymer flooding-binary compound flooding multi-development stage was obtained.

## 3. Variation Rule of Dimensionless Productivity Index: Case Study of Block W of JZ9-3 Oilfield

### 3.1. Oilfield Overview

The basic parameters of the W Block of JZ9-3 oilfield are provided in Table 1.

**Table 1.** Basic data of the JZ9-3 oilfield.

| Parameter | Value |
|---|---|
| Porosity (%) | 0.29 |
| Well spacing (m) | 350 |
| Permeability ($\mu m^2$) | 1.50 |
| Capillary pressure (Pa) | 0.20 |
| Formation pressure (MPa) | 16.43 |
| Water injection pore volume multiple (PV) | 0.64 |
| Polycondensation pore volume multiple (PV) | 1.00 |
| Note binary pore volume multiple (PV) | 1.60 |
| Ground water viscosity (mPa·s) | 0.80 |
| Underground crude oil viscosity (mPa·s) | 7.00 |
| Polymer viscosity (mPa·s) | 9.89 |
| Polymer–surfactant binary compound viscosity (mPa·s) | 12.40 |

Excluding the influence of production factors, production well W1 was selected as the target well in this study, and the change curves of the moisture content and the daily fluid volume in the three phases of water flooding, polymer flooding, and binary compound flooding were obtained, as shown in Figure 2.

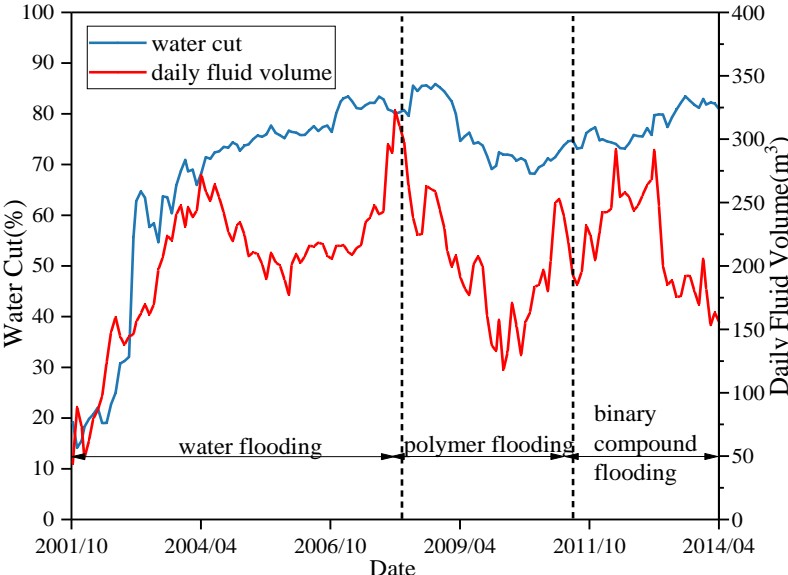

**Figure 2.** Variation in water cut and daily oil production of well W1.

Figure 2 shows that in the stage of water flooding development, the moisture content in well W1 first rises rapidly, and then rises slightly with an average increase of 82.15% monthly. In the early stage of water injection, the daily liquid production yield is significantly increased; in the middle stage of water injection, a precipitation funnel formed. Because of the decrease in daily water production, the daily liquid production yield also decreased a little. At the end of water injection, because of the formation of an advantageous channel, a large amount of injected water was produced, resulting in a sharp rise in the daily liquid production yield, with an average increase of 82.28% monthly. After polymer injection, the moisture content decreased slowly and then increased slightly. The decrease indicates that well W1 is suitable for polymers. However, the implementation time of polymer flooding is early and the moisture content decreases less than that of water flooding stage. The fluctuation in the moisture content is caused by high reservoir pressure. The overall trend in the moisture content showed the average decline rate of 12.76% monthly. The daily liquid production yield showed a trend of sharp rising after a sharp drop, and the overall trend was also downward, with an average decline rate of 31.74% monthly. After the injection of surfactant, the moisture content soon decreased slightly, and then the moisture content slowly fluctuate and increase, with an average increase rate of 8.07%, indicating that the binary compound system is not efficient. The daily liquid production yield fluctuated significantly and decreased, with an average decline rate of 64.57% by August 2014.

According to our analysis of the JZ9-3 oilfield, the liquid yield in the chemical flooding stage decreased significantly, the water content remained basically unchanged, and the corresponding oil production decreased, so it was necessary to research the change law of liquid content in the chemical flooding process of this oilfield. The dimensionless productivity index can describe the whole process of oilfield development and the influence of production pressure difference can be ignored, which has certain practical significance for offshore oilfields where chemical flooding has been conducted and bottom-hole flow pressure is difficult to measure. In this study, dimensionless liquid productivity index was used as the index of liquid production decline to evaluate and guide the liquid production situation of this oilfield [10].

### 3.2. Comparative Analysis of the Dimensionless Productivity Index

The dimensionless productivity index model established in this paper was used to calculate the liquid productivity index in this W Block, and the calculation model established was compared with the dimensionless productivity index calculated from the actual statistical data of the mine. The comparison results are shown in Figure 3. The dimensionless productivity index changes with the

saddle type and the PV number injected, and the results produced by the model are generally lower than the actual statistics.

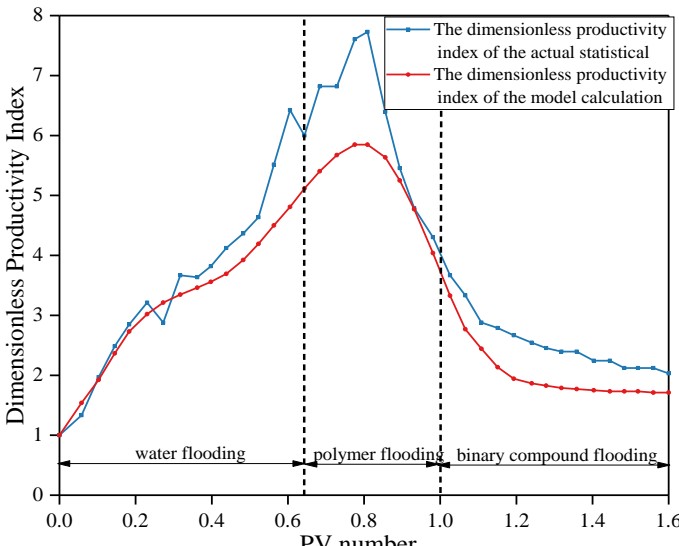

**Figure 3.** Comparison of the change in the dimensionless productivity index of the actual statistics and the new calculation model with the pore volume (PV) number injected.

From a whole development perspective, the curve of water flooding stage presents an upward trend. The dimensionless productivity index obtained using the calculation model in this paper is basically consistent with the actual situation, with an average error rate of 4.67%.

In the polymer flooding stage, the calculation model deviates considerably from the actual statistical data. After the polymer injection effect, the dimensionless productivity index has a monotonic decline, indicating that the oil displacement effect of the reservoir improved, with an average error rate of 17.65%.

When the injection PV number reaches 1.0, the binary composite system is injected, and the dimensionless liquid yield index first decreases to a certain extent, and then gradually stabilizes. The actual decline rate was 50%, and the decline rate obtained by the calculation model was 52.63%, with an average error rate of 18.50%. The error rate in the injection binary stage is higher than in the injection polymerization stage and water injection stage, which may be due to the emulsification induced by the introduction of the surfactant, resulting in the increase in the fluid flow resistance [22].

The calculation results of the model are in good agreement with the actual results, and the average calculation error of the two curves was about 10.38%. We proved that the error rate of the calculation model of the dimensionless productivity index established in this study is within the allowable range, and can be used to evaluate various stages of oilfield productivity.

### 3.3. Factors Affecting of the Dimensionless Producyivity Index

The calculation method in this paper is based on the consideration of the factors influencing the dimensionless productivity index, including water saturation, oil–water viscosity ratio, oil–water relative permeability, and pressure gradient, so the method can more accurately reflect the changing trend in reservoir fluid production. Since water saturation and water–oil viscosity ratio have a greater impact on the dimensionless productivity index, the following research was conducted on the influence of water saturation and oil–water viscosity ratio on the dimensionless productivity index.

### 3.3.1. Water Saturation

Using the dimensionless productivity index calculation model established in this paper, with Equations (18), (21), and (24), we calculated the dimensionless productivity index of the target

production wells in the stages of water flooding, polymer flooding, and dual flooding in block W. Under different water saturations, the relationship between the dimensionless productivity index and the PV number was determined (Figure 4). At the same injection PV number, as the water saturation increases, the change range of the dimensionless productivity index is more drastic.

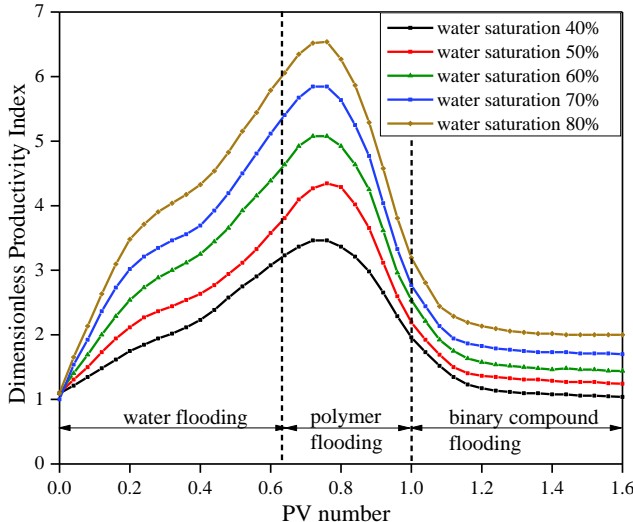

**Figure 4.** The variation in the dimensionless productivity index with the PV number injected under different water saturations.

Figure 4 shows that in the water flooding stage, the dimensionless productivity index increases gradually and the average growth rate of the index increased by 7.81/PV. The reason for this is that when the water saturation is low, the oil is mainly transported in the formation, and the average viscosity of the fluid is higher and decreases slowly. As the injected PV number increases, the water saturation in the formation gradually increased, and the average viscosity of the formation fluid gradually decreased. As the reservoir liquid production rate increased, the dimensionless productivity index gradually increased. When water saturation was 40%, 50%, 60%, 70%, and 80%, the increase range of the dimensionless productivity index was 68.54%, 69.93%, 71.72%, 72.46%, and 74.45% respectively.

In the ineffective polymer flooding stage, with the increase in the number of injected PVs, the dimensionless productivity index first maintained an upward trend for a period of time. When the water saturation was 40%, 50%, 60%, 70%, and 80%, the increase range of the dimensionless productivity index was 5.64%, 6.02%, 6.28%, 6.51%, and 6.75%, respectively. When the PV number increased to 0.78, the dimensionless productivity index peaked. After the polymer injection was effective, with an increase in the number of injected PVs, the dimensionless productivity index decreased rapidly. When water saturation was 40%, 50%, 60%, 70%, and 80%, the decrease range of the dimensionless productivity index was 61.74%, 62.54%, 63.72%, 64.74%, and 65.46%, respectively. The dimensionless productivity index showed an ascending section because the seepage resistance at the initial stage of the polymer injection continued the downward trend of the water flooding stage; the dimensionless productivity index decreased significantly because the polymer solution was a non-Newtonian fluid with high viscosity. By injecting the polymer, the leading edge gradually formed an oil-rich zone. When the oil-rich zone entered the control zone but had not yet broken through the well, the heavy oil saturation increased and the oil-rich zone formed by the polymer flooding leading edge had a lower fluidity. The reservoir seepage resistance increased rapidly and the moisture content decreased slowly, resulting in a significant decrease in the liquid production capacity of the production well in the polymer flooding stage compared with the water flooding stage.

In the binary compound injection stage, the dimensionless productivity index was generally further reduced; when water saturation was 40%, 50%, 60%, 70%, and 80%, the decrease range of the dimensionless productivity index was 71.84%, 42.08%, 42.22%, 42.46%, and 42.95%, respectively.

Then, the dimensionless productivity index gradually stabilized because after the breakthrough in the oil-rich zone, the oil saturation in the control zone was higher, the seepage resistance gradually reached the maximum and maintained a high level, and the corresponding liquid production index remained stable.

### 3.3.2. Water–Oil Viscosity Ratio

The dimensionless productivity indexes of the reservoir development stage with different water–oil viscosity ratio were calculated using the calculation model established in this paper. The calculation results show that the higher the oil–water viscosity ratio, the higher the dimensionless productivity index.

As the injected PV number increased, in the stage of water flooding, when the water–oil viscosity ratio is 5, 10, 20, 40, and 80, the increase range of the dimensionless productivity index is 0%, 34.44%, 66.67%, 82.14%, and 88.10%, respectively. In the ineffective polymer flooding stage, when water–oil viscosity ratio is 5, 10, 20, 40, and 80, the increase range of the dimensionless productivity index is 49.75%, 47.28%, 43.33%, 38.89%, and 32.26%, respectively. After the polymer injection is effective, when the water–oil viscosity ratio is 5, 10, 20, 40 and 80, the decrease range of the dimensionless productivity index is 43.63%, 48.55%, 58.21%, 58.59%, and 67.74%, respectively. In the binary compound injection stage, when the water–oil viscosity ratio is 5, 10, 20, 40, and 80, the decrease range of the dimensionless productivity index is 49.15%, 62.22%, 68.72%, 70.46%, and 75.45%, respectively.

We concluded from the curve characteristics in Figure 5 that during the development process of water flooding, polymer flooding, and binary compound flooding in the reservoir, the dimensionless productivity index changes with the different viscosity ratios in two stages. The high water–oil viscosity ratio showed an upward trend during the period of water flooding and polymerization without effect, and the increase range gradually increased. The low water–oil viscosity ratio decreases slightly during the stage of water flooding and quickly rises during the period of no-effect polymer injection. The low water–oil viscosity ratio decreases slightly in the water flooding stage and quickly rises during the period of no-effect polymer injection. In the chemical flooding stage after the polymer injection is effective, all ratios present a downward trend, and the decline gradually decreases, and finally tends to be stable.

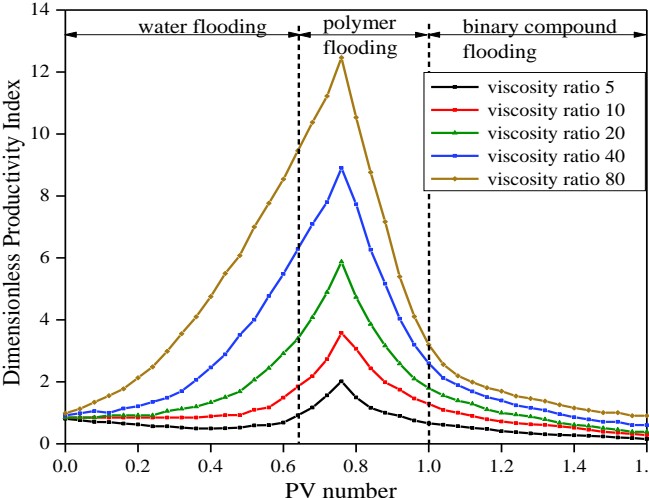

**Figure 5.** The variation in the dimensionless productivity index with the PV number injected under different water–oil viscosity ratios.

In reservoirs with high and low water–oil viscosity ratios, the variational trends in the dimensionless productivity index with PV number injected are inconsistent, which reflects the influence of oil–water phase permeability. In reservoirs with low water–oil viscosity ratios, the resistance increases

gradually because of the rapid decline in oil-phase permeability, so the dimensionless recovery index displays a small and slow declining trend. During the period of no-effect polymer injection, the permeability of the water phase increases significantly and the conductivity capacity of water increases accordingly. The water–oil viscosity ratio has a stronger impact on the total resistance of oil and water in the formation, and the total seepage resistance begins to decline and the dimensionless recovery index is higher.

## 4. Conclusions

In this paper, based on Darcy's Law and the theory of non-piston leading edge propulsion, we considered the influence of capillary pressure and derived a differential equation for leading edge propulsion distance. And the following conclusions are obtained through specific analysis based on actual data:

- In this study, we established a calculation model of the dimensionless productivity index suitable for the multiple development phases of oilfields, including water flooding, polymer flooding, and binary compound flooding, and we established an index suitable for medium and low permeability reservoirs. The calculation results of the dimensionless productivity index model were compared with the actual statistical results; the calculation error rate of the dimensionless productivity index for the three oilfield development phases were 4.67%, 17.65%, and 18.50%, respectively. The average error rate was 10.38% in the overall development phase.
- In the development processes of water flooding, polymer flooding, and binary compound flooding, the dimensionless productivity index shows a trend of first rising, then falling, and finally stabilizing with the increase in PV injection. Before the effects of polymer injection occur, the dimensionless productivity index increased steadily and the increase rate was 71.75% monthly, and the oil displacement effect of the reservoir worsened. After the polymer injection became effective, the dimensionless productivity index steadily decreased, the decrease rate was 52.63% monthly, and the effect of oil displacement of the reservoir improved. After injection into the binary compound system, the dimensionless productivity index further decreased at a rate of 41.65% monthly, and the oil displacement effect of the reservoir was further improved. The method can be used to evaluate the law of the production fluid of the oil well after the polymer injection and binary compound injection.
- Under constant pressure conditions, the dimensionless productivity index is positively and linearly related with water saturation; under the same PV number, the dimensionless productivity index was larger, and the water saturation was higher. The change in the dimensionless productivity index was more drastic. During the stages of water flooding, ineffective polymer flooding, effective polymer flooding, and binary compound flooding, the average increase, and decrease ranges of the dimensionless productivity index were 71.42%, 6.24%, −63.64%, and −42.31%, respectively.
- Under constant pressure conditions, the dimensionless productivity index has a positively correlated linear relationship with the water–oil viscosity ratio; under the same PV number, the dimensionless productivity index was larger, and the water–oil viscosity ratio was higher. In the stages of water flooding, ineffective polymer flooding, effective polymer flooding, and binary compound flooding, the average increases and decreases of the dimensionless productivity index were 54.27%, 42.31%, −55.34%, and −65.20%, respectively.

**Author Contributions:** B.H. and Q.L. proposed the computational model and wrote the paper. C.F., C.H., and K.S. contributed to the analysis of the results. All the authors reviewed the manuscript.

**Funding:** This work was financially supported by Sub-projects of major national projects (016ZX05025-003-05); National Natural Science Foundation of China (51804077).

**Conflicts of Interest:** The authors declare no conflict of interest.

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
