# Peer review of "A Calculation Model of the Dimensionless Productivity Index Based on Non-Piston Leading Edge Propulsion Theory in Multiple Oilfield Development Phases"

_processes, doi:10.3390/pr7110821_

Round 1

Reviewer 1 Report

Nice piece of work, it is well presented and edited. The list of references definitely needs to be updated. given the fact that this is a highly technical article, you should update the reference list and include more recent (and relevant) articles to support your work. As it stands now there are only 4/25 that come from the last 5-6 years. You need to include more modern research work and you might consider articles like, ‘ISS Stability Control: A Comparison of Different Approaches’, International Journal of Engineering, Applied Sciences and Technology, (ISSN: 2455-2143), Vol. 2, Issue 8, December 2017

Author Response

Response to Reviewer 1 Comments:

First of all, thank you very much for your comments on this article. We tried our best to improve the manuscript according to your comments and made some changes in the manuscript. And the replies to your comments are as follows. We appreciate for your warm work earnestly, and hope that the correction will meet with approval. Once again, thank you very much for your comments and suggestions.

Point 1: Nice piece of work, it is well presented and edited. The list of references definitely needs to be updated. given the fact that this is a highly technical article, you should update the reference list and include more recent (and relevant) articles to support your work. As it stands now there are only 4/25 that come from the last 5-6 years. You need to include more modern research work and you might consider articles like, ‘ISS Stability Control: A Comparison of Different Approaches’, International Journal of Engineering, Applied Sciences and Technology, (ISSN: 2455-2143), Vol. 2, Issue 8, December 2017

Response 1: Special thanks to the article you suggested and we have read them. It is really true as you suggested that the article you proposed is a good discussion of comparative methods. We have added the reference and the content. The revised content are as follows:

Michail Papoutsidakis proposed ISS Stability Control, and how can the method be utilized by the scientifically-oriented Engineer, either in human measures or in production, whereas an interdisciplinary role in evaluation of oilfield production [19].

Reference: Michail Papoutsidakis. ISS Stability Control: A Comparison of Different Approaches. International Journal of Engineering, Applied Sciences and Technology. 2017, 2(8), 2455-2143.

We have tried our best to revise our manuscript according to the comments. We would like to express our great appreciation to you for comments on our paper. Looking forward to hearing from you.

Thank you and best regards.

Reviewer 2 Report

Very interesting theoretical work with elements of practical research.
The presented solution should find a group of companies interested in implementation in their plants.

Author Response

Response to Reviewer 2 Comments:

First of all, thank you very much for your comments on this article. We tried our best to improve the manuscript according to your comments and made some changes in the manuscript. And the replies to your comments are as follows. We appreciate for your warm work earnestly, and hope that the correction will meet with approval. Once again, thank you very much for your comments and suggestions.

Point 2: Very interesting theoretical work with elements of practical research.

The presented solution should find a group of companies interested in implementation in their plants.

Response 2: Special thanks to the problems which you proposed. It is really true as you suggested. The non-production liquid index we calculated for evaluating the oil field production has been applied to JZ9-3 oil field, and we also hope to apply it to more oil field production evaluation.

We have tried our best to revise our manuscript according to the comments. We would like to express our great appreciation to you for comments on our paper. Looking forward to hearing from you.

Thank you and best regards.

Reviewer 3 Report

Dear authors,

Please find my commetns in attached PDF.

Regards,

Reviewer (HR)

Author Response

Response to Reviewer 3 Comments:

First of all, thank you very much for your comments on this article. We tried our best to improve the manuscript according to your comments and made some changes in the manuscript. And the replies to your comments are as follows. We appreciate for your warm work earnestly, and hope that the correction will meet with approval. Once again, thank you very much for your comments and suggestions.

Point 1: Make the same in entire text / name + surname or initial + surname.

Response 1: Special thanks to the problems which you proposed. In the article, I changed the format of initial + surname. Here we did not list the changes but marked in red in revised paper.

Point 2: et al. and format issue

Response 2: Special thanks to the problems which you proposed. Here we did not list the changes but marked in red in revised paper.

Point 3: The word "formation" has unique lithostratigraphic meaning. Use here words "rock" or "reservoir".

Response 3: Special thanks to the problems which you proposed. It is really true as you suggested that the word is inappropriate. We have corrected to use the word “reservoir”.

Point 4:  the W block of the JZ9-3 oilfield

Readers need to know:

a) location of the field, b) type and age of reservoir, c) geological evolution of field structure, d) production history and life span of the field.

Response 4: Special thanks to the problems which you proposed. It is really true as you suggested that listing the main factors will help the reader get ready for when we introduce the indexes later on in the paper. I have described these four factors in the article and made a reasonable adjustment in the order of the article.

Point 5: fwmoisture content, %

Define very precise what is moisture content in reservoir, how it can be distinguished from wettability and saturation!

Response 5: Special thanks to the problems which you proposed. We have consulted materials to determine the specific meaning of this symbol, which is clearly defined as water cut.

Point 6: output = recovering

could indicate changes in So from oil/water toward oil zone?

Response 6: Special thanks to the problems which you proposed. It is our negligence and we are sorry about this. We have made correction according to your comments. We modified the three diagrams of Figure1, and hope could indicate changes in So from oil/water toward oil zone.

(a).water flooding stage.   (b).polymer flooding stage.  (c).binary compound flooding stage.

Figure 1. Schematic diagram of the leading edge propulsion of the water flooding-polymer flooding-binary compound flooding in multi-development phases

Point 7: Since gravity has little effect on the dimensionless productivity index, the influence of

gravity is ignored in this paper.

Without gravity (a) migration never happened, (b) water flooding would not work.

Response 7: Special thanks to the problems which you proposed. We are deeply sorry for our negligence in not explaining the “gravity”. In the absence of gravity, the leading edge of the finger can not be pushed, but because the capillary force will appear arcuate leading edge. At the same time, this paper considers artificial water flooding, which can be used as power source. We have modified the three diagrams of Figure1.

Point 8: vw—water phase seepage velocity, m/s;

vo—oil phase percolation velocity, m/s

migration, moving, but seepage???

Response 8: Special thanks to the problems which you proposed. We have determined the correct way of expression as follows:

vw—water phase migration velocity, m/s;

vo—oil phase migration velocity, m/s

Point 9: the permeability resistance of formation fluid

Be precise. Permeability of rock is enough high that such mixture can flow through it.

Permeability is not resistance, it is physical value of availability to flow.

Response 9: Special thanks to the problems which you proposed. We have determined the correct way of expression as follows:

the migration resistance of formation fluid

Point 10: Binary compound flooding stage.

Title???

Response 10: Special thanks to the problems which you proposed. This part is a title. We hope you can point out the problem more clearly if you still think the title has something wrong.

Point 11: the oil–water phase permeability,

rock permeability, but oil/water wettability.

Response 11: Special thanks to the problems which you proposed. We have corrected it as you suggested.

Point 12: The variation rule of the dimensionless productivity index -- Take the W Block of the
JZ9-3 oilfield As A Case

Case study in the Block A...

Response 12: Special thanks to the problems which you proposed. We have modified it as follows:

Case study in the Block W of the JZ9-3 oilfield

Point 13: The JZ9-3 oilfield belongs to the offshore heavy oilfield.

Oilfield does not belong to anything, but reservoir is saturated with heavy oil

Response 13: Special thanks to the problems which you proposed. We have modified it as follows:

The JZ9-3 reservoir belongs to the offshore heavy reservoir.

Point 14: The basic data of the W Block of JZ9-3 oilfield

This is for the first part of the paper, not here.

Response 14: Special thanks to the problems which you proposed. We have adjusted the content sequence of the article appropriately and moved the introduction of JZ9-3 to the first part.

Point 15: moisture content

Is this water portion in produced fluid? Sw?

Response 15: Special thanks to the problems which you proposed. We have modified it as water cut. Meanwhile, the vertical coordinate and legend in Figure2 have also been changed to water cut.

Point 16: Figure3.

Some texts of figure(s) are too tiny.

Response 16: Special thanks to the problems which you proposed. We have adjusted the size of some texts of figures of Figure1, Figure2 and Figure3.

Point 17: 2.3 Factors Affecting of the Dimensionless Producyivity Index

mostly describe the figures with lot of sentences (what is visible by itself), but do not lead the reader toward the most important conclusions that could be derived from graphs.

I felt that water injection gave temporary improvement, and later polymers did the same.

But cannot point out the clear figure for percentage of improvement and lasting of such "sunshine" periods.

Response 17: Special thanks to the problems which you proposed. Your suggestion is very correct, and I have added additional explanation to it, hoping to guide readers to a clearer conclusion. Here we did not list the changes but marked in red in revised paper.

At the same time, I think dimensionless productivity index is a production index used to evaluate the production situation of the oil field, which can play a certain reference value. However, we have not considered the improvement of the specific production situation of the oil field.

Point 18: Conclusion

This is review of results. But what you concluded? Your index will work in sandstone reservoirs with heavy oil or something like that...?

Response 18: Special thanks to the problems which you proposed. We've limited the scope of the index to the following as follows:

This paper establishes a calculation model of the dimensionless productivity index suitable for the multiple development phase of oilfield, including water flooding, polymer flooding, and binary compound flooding. And the index suitable for medium and low permeability reservoir.

Point 19: in three oilfield multiple development phases

Three? But in the beginning of the paper is mentioned only one.

Response 19: Special thanks to the problems which you proposed. We have made correction according to your comments as follows:

in three development phases of oilfield.

Point 20:  In the development process of water flooding, polymer flooding and binary compound flooding, the dimensionless productivity index shows a trend of first rising, then falling and finally stabilizing with the injection of PV.

Numbers

Response 20: Special thanks to the problems which you proposed. We have modified it as follows:

In the development process of water flooding, polymer flooding and binary compound flooding, the dimensionless productivity index shows a trend of first rising, then falling and finally stabilizing with the injection of PV. Before the effect of polymer injection, the dimensionless productivity index increased steadily, and the increase rate was 71.75%, and the oil displacement effect of the reservoir worsened. After the effect of polymer injection, the dimensionless productivity index steadily decreased, and the decrease rate was 52.63%, and the effect of oil displacement of the reservoir was improved. After injection into the binary compound system, the dimensionless productivity index was further reduced and the decrease rate was 41.65%, and the oil displacement effect of the reservoir was further improved. The method can be used to evaluate the law of the production fluid of the oil well after polymer injection and binary compound injection.

Point 21: Under constant pressure conditions, the dimensionless productivity index had a positively correlated linear relationship with the water saturation; under the same PV number, the dimensionless productivity index was larger, and the water saturation was higher.

Numbers

Response 21: Special thanks to the problems which you proposed. We have modified it as follows:

Under constant pressure conditions, the dimensionless productivity index had a positively correlated linear relationship with the water saturation; under the same PV number, the dimensionless productivity index was larger, and the water saturation was higher. The change in the dimensionless productivity index was more drastic. In the stage of water flooding, ineffective polymer flooding, effective polymer flooding, and binary compound flooding, the average increase and decrease range of the dimensionless productivity index were 71.42%, 6.24%, -63.64% and -42.31% respectively.

Point 22: Under constant pressure conditions, the dimensionless productivity index had a positively correlated linear relationship with the water–oil viscosity ratio.; under the same PV number, the dimensionless productivity index was larger, and the water–oil viscosity ratio was higher.

Numbers

Response 22: Special thanks to the problems which you proposed. We have modified it as follows:

Under constant pressure conditions, the dimensionless productivity index had a positively correlated linear relationship with the water–oil viscosity ratio.; under the same PV number, the dimensionless productivity index was larger, and the water–oil viscosity ratio was higher. In the stage of water flooding, ineffective polymer flooding, effective polymer flooding, and binary compound flooding, the average increase and decrease of the dimensionless productivity index were 54.27%, 42.31%, -55.34% and -65.20% respectively.

We have tried our best to revise our manuscript according to the comments. We would like to express our great appreciation to you for comments on our paper. Looking forward to hearing from you.

Thank you and best regards.

Round 2

Reviewer 3 Report

Dear authors,

You made great improvement and wrote rarely polite and comprehensive letter to editor. I accepted your paper and propose editor to publish it, with great pleasure.

Wish you all the best in future researching,

Reviewer 3